# Relationship between Nutritional Status, Body Composition, Muscle Strength, and Functional Recovery in Patients with Proximal Femur Fracture

**DOI:** 10.3390/nu14112298

**Published:** 2022-05-30

**Authors:** Hiroshi Irisawa, Takashi Mizushima

**Affiliations:** Department of Rehabilitation Medicine, Dokkyo Medical University, Mibu 3210293, Japan; mizusima@dokkyomed.ac.jp

**Keywords:** body composition, rehabilitation, sarcopenia, proximal femur fractures, ADL, elderly

## Abstract

Sarcopenia is a major issue among the elderly. However, the effects of nutritional status and body composition on functional recovery in patients with proximal femur fractures (PFF) remain unclear. Hence, this study aimed to investigate the effects of nutritional status, body composition (skeletal muscle mass and muscle quality measured by phase angle [PhA] values), and muscle strength on the improvement in activities of daily living (ADL) in patients with PFF. We enrolled patients with PFF admitted to a rehabilitation unit. Nutritional status, body composition, grip strength, and motor Functional Independence Measure (FIM) score were assessed on admission day and at 4 weeks thereafter. Of 148 patients, 84 had femoral neck fractures, and 64 had trochanteric fractures. The mean motor FIM score was 49.2 points at admission and 64.9 points after 4 weeks. In multivariate analysis, higher geriatric nutritional risk index and PhA measured by anthropometry were associated with a significantly higher FIM score after 4 weeks. Muscle strength and quality changes significantly correlated with ADL improvement. Poor nutritional status and decreased muscle strength and quality interfered with ADL recovery. Nutritional management before injury and from the acute phase, and rehabilitation to maintain skeletal muscle status, are important for ADL recovery.

## 1. Introduction

Post proximal femur fracture (PFF) muscle weakness will be partially eased by rehabilitation [1]. To evaluate muscle, muscle strength has been the most useful parameter. However, muscle mass and muscle quality have also become increasingly important parameters to assess muscle function [2,3,4]. To the best of our knowledge, no studies have examined whether recovery of muscle strength in post-PFF rehabilitation is accompanied by recovery of muscle mass or muscle quality.

Lower-limb muscle strength has been related to walking ability and activities of daily living (ADL) status in patients with post-PFF [5]. Skeletal muscle mass is related to muscle strength, and a strong correlation exists between muscle fiber diameter and tension in humans [6]. Lower-limb skeletal muscle mass is also related to muscle strength in patients with PFF [7,8]. Conversely, recent studies have found that muscle mass only moderately correlates with muscle strength in the elderly; clearly, muscle weakness cannot be explained by loss of muscle mass alone [9,10,11]. This observation is attributable to the presence of extracellular fat and extracellular fluid in the skeletal muscle tissue. Moreover, intermuscular fat accumulation is responsible for reduced muscle strength [12]. Therefore, muscle quality and muscle mass should both be considered in the assessment of muscle strength. Muscle quality is commonly assessed by CT, MRI, and ultrasound [13]; however, phase angle (PhA), which is measured by body composition monitors, reflects muscle quality. The European Working Group on Sarcopenia in Older People (EWGSOP) 2019 consensus statement indicated that PhA could be an index of overall muscle quality [14].

In addition, a poor nutritional status inhibits physical function recovery in patients with post-PFF [15]. Furthermore, most patients with PPF are elderly. Elderly patients have reduced muscle strength, muscle mass, and balance capacity due to sarcopenia [16]. Therefore, nutritional assessment and nutritional intervention are important in patients undergoing stroke rehabilitation, considering that malnutrition results in decreased physical function through skeletal muscle loss. The geriatric nutritional risk index (GNRI) is a very simple and objective method based on body weight, height, and serum albumin levels that is used to assess the nutritional status in numerous pathological conditions [17]. As mentioned, assessing the skeletal muscle mass and nutritional status is important in PFF rehabilitation; however, their effects on ADL recovery remain unclear. Hence, this study aimed to clarify the effects of muscle strength, mass, and quality and nutritional status on ADL recovery in patients rehabilitating from PFF.

## 2. Materials and Methods

This prospective study was conducted at two rehabilitation units in Japan between January 2017 and June 2018. All patients provided written informed consent before enrollment. The study conformed to the Declaration of Helsinki. The ethics committee of the Setagaya Memorial Hospital approved the study protocol (H30-003). The study initially included 157 consecutive patients with PFF. Patients with a pacemaker, high ADL score (motor FIM items > 81), severe cognitive impairment, severe dysphasia, and early discharge were excluded (Figure 1).

### 2.1. Nutritional Status

According to Wakabayashi [16], the meal consumption of each patient during hospitalization was 1500–2000 kcal (protein, 1.5 g/kg/day) and was managed by the dietitian. We assessed the nutritional status by calculating the GNRI on admission, as described by Bouillanne et al. [18]. The GNRI is a universally adopted tool for evaluating one’s nutritional status. It is an effective and simple risk index that evaluates a patient’s nutritional risk and is also a proven predictive index for the prognosis of the elderly, patients with dialysis and cardiovascular disease, and health care. The nutritional status of each patient was evaluated using the following GNRI formula: GNRI = (1.489 × albumin [g/L]) + (41.7 × [weight/WLo]), where WLo denotes ideal weight and was calculated using the Lorentz equation (for males: H − 100 − [(H − 150)/4]; for females: H − 100 − [(H − 150)/2.5]; H: height).

### 2.2. Bioelectrical Impedance Analysis (BIA)

We used the InBody S-10 analyzer (InBody Japan, Tokyo, Japan), which applies a 200 μA current at frequencies of 5, 50, and 250 kHz after 10 min of rest at ambient temperature. Immediately after admission, all patients underwent BIA. For 2 h before the measurements, the patients did not consume any liquids or solids. The same operator performed the analysis in all patients. For BIA, the areas chosen for electrode placement were shaved (if needed) and cleaned before attaching one electrode on all four limbs of each patient while lying supine. The patient’s weight was measured while lying on a stretcher, and the empty stretcher weight was subtracted from the total weight. Skeletal muscle mass and PhA were then measured. The whole-body PhA at 50 kHz was calculated from the impedance values. The skeletal muscle mass index (SMI) [19] was used to standardize the muscle mass values.

### 2.3. Muscle Strength Assessment

A dynamometer is a quick, convenient, and low-cost tool used to clinically assess overall muscle strength [20]. We used a dynamometer (Grip A TKK5001; Takei Scientific Instruments Co., Ltd., Niigata, Japan) to assess grip strength according to the recommendations of the American Society of Hand Therapy [21]. Examinations were performed bilaterally on the left and right sides. The patient sat in a chair (with backrest, without armrests) with the lower limbs on the ground. The examiner verbally encouraged the patient to exert maximum effort in performing the test. The test was repeated three times on each side and the average value was recorded.

### 2.4. Functional Measurements

The ADL status was assessed by motor Functional Independence Measure (FIM) scoring. The FIM contains 13 items related to motor tasks, each rated on a 7-point ordinal scale; higher scores indicate greater independence [22]. This scale is used mainly during neurological rehabilitation (including patients with stroke and brain injury) and geriatric rehabilitation [23]. FIM was assessed by members of the multidisciplinary rehabilitation team. FIM, BIA, nutritional status, and muscle strength assessments were performed on all patients on the day of admission and 4 weeks later. The amount of change in the motor FIM score over 4 weeks was used as a marker of functional recovery.

### 2.5. Statistical Analysis

Continuous variables are expressed as mean ± standard deviation (SD). Differences between male and female patients were assessed using independent t-test. Furthermore, *p* values of less than 0.05 were considered statistically significant.

The relationships between malnutrition absence (GNRI > 92), high SMI (male > 7.0 kg/m^2^, female > 5.4 kg/m^2^), high PhA (male > 3.5°, female > 3.0°), high grip strength (male > 26.0 kgw, female > 18.0 kgw), and functional recovery were estimated using odds ratios and 95% confidence intervals obtained from multivariate logistic regression models. Considering that SMI, body fat mass, and PhA differ between sexes, the analysis was conducted on a sex basis [24]. We set the cutoff values of body fat percentage, SMI, and grip strength according to the criteria for the older Japanese population [25] and Asian Working Group for Sarcopenia (AWGS) [26]. All data were statistically analyzed using IBM SPSS Statistics version 25 (IBM Corp., Armonk, NY, USA).

## 3. Results

### 3.1. Descriptive Characteristics

In total, 148 patients (121 females and 27 males; 84 had femoral neck fractures, 64 had trochanteric fractures; mean age: 84.8 years) were included. Table 1 summarizes the descriptive and functional characteristics of the patients. All patients were Japanese (Asian).

The mean duration from PFF onset to rehabilitation unit admission was 22.6 days. All patients spent approximately 140 min per day on the rehabilitation program. Males were significantly taller (*p* < 0.05) and heavier (*p* < 0.05) than females; however, they did not significantly differ in terms of age or body mass index (BMI) (Table 2).

Males also had higher muscle strength, muscle quality, and SMI than females. The participants suffered from PFF, so walking speed could not be measured. Therefore, none of them met the diagnostic criteria for sarcopenia (AWGS) [26]. However, 33 of them exceeded the AWGS criteria for SMI and grip strength. At 4 weeks, both males and females significantly improved in nutritional status, motor FIM scores, muscle strength, and muscle quality (all, *p* < 0.05). However, SMI decreased during 4 weeks in males and females (Table 3 and Table 4).

### 3.2. Univariate Analyses

We investigated which covariates were associated with functional recovery. In the univariate analysis, malnutrition absence and a high muscle quality on admission were associated with functional recovery (Table 5).

We also investigated ADL recovery of patients and assessed its correlation with nutritional status, muscle mass, muscle strength, and muscle quality. ADL recovery significantly correlated with muscle strength and muscle quality (male: r = 0.41 and 0.39, female: r = 0.43 and 0.52, respectively), and the correlation was stronger in males. Meanwhile, ADL recovery almost had no correlation with SMI and nutrition status (male: r = −0.03, −0.01, female: r = −0.02, 0.03, respectively) (Figure 2A–D and Figure 3A–D).

The vertical axis represents the change in ADL within 4 weeks, and the horizontal axis represents the changes in nutritional status, grip strength, muscle quality, and SMI, respectively. ADL recovery significantly correlated with muscle strength and muscle quality (r = 0.41 and 0.39, respectively), but did not correlate with SMI and nutritional status (r = −0.03 and −0.02, respectively).

The vertical axis represents the changes in ADL within 4 weeks, and the horizontal axis represents the changes in nutritional status, grip strength, muscle quality, and SMI, respectively. ADL recovery significantly correlated with muscle strength and muscle quality (r = 0.43 and 0.52, respectively). The correlation was stronger in males than in females. However, ADL recovery showed no correlation with SMI and nutritional status (r = −0.01 and 0.03, respectively).

## 4. Discussion

To the best of our knowledge, this is the first study to clarify the relationships between BIA, nutritional status, and functional recovery in patients with PFF. At the start of intensive fracture rehabilitation, nutritional status and muscle quality already considerably influenced functional recovery.

Patients with fractures have poor nutritional status, muscle weakness, and muscle mass loss [1]. In addition, such patients have difficulty recovering their ADL after fracture [27]. In the elderly, malnutrition is a factor that worsens the prognosis [15].

### 4.1. Nutritional Status

In this study, we used the GNRI for assessing nutritional status. As reported by Bouillanne et al. [18], the GNRI is an indicator of nutritional status, and a value of 98 or higher indicates well-nourished. The mean GNRI of our patients was below 98, and both male and female patients had malnutrition. Malnutrition is common after PFF, and acute nutritional status affects functional recovery [15]. The present study evaluated the nutritional status of patients with PFF in the rehabilitation unit. We found that the nutritional status of the patients clearly affected their functional recovery. Malnutrition is generally caused by starvation, and acute or chronic illness [28], and patients undergoing rehabilitation may have any of these conditions. The elderly are often malnourished after acute treatment or chronic diseases such as chronic renal failure. These conditions can cause malnutrition. The elderly require more protein than younger people. Decreased protein intake leads to decreased muscle mass [29], but the combination of protein intake and exercise therapy increases muscle mass in the elderly [30]. In the present study, patients with PFF exhibiting a low GNRI had poor functional recovery after rehabilitation. Therefore, the evaluation of muscle mass related to functional recovery and nutrition intervention is necessary. However, patients with PFF manifesting cachexia can possibly have a low GNRI. However, the effect of rehabilitation on cachexia is still unknown [31]. Additionally, their mobility may have been poor because the positive effects of rehabilitation are impossible to obtain from these patients.

### 4.2. Body Composition

The simplest way to measure skeletal muscle mass is to assess limb circumference. Local muscle mass is usually measured by CT, magnetic resonance imaging, and ultrasound. In addition, muscles throughout the body are generally measured by dual-energy X-ray absorption and BIA.

The BIA applies a weak electric current to the body and indirectly measures water content, body fat content, and muscle mass from its electrical impedance. BIA is noninvasive and simple, but measurement results are easily affected by changes in conductivity due to body fluid levels, such as dehydration and edema, and body temperature [32]. Considering these factors, the BIA method provides sufficiently reliable results compared to the DXA method. The BIA method is also superior to the DXA method in that there is no radiation exposure [33].

SMI, muscle strength, and PhA have been reported to differ between males and females [25,34]. Therefore, our analysis was performed separately for male and female patients. Both groups showed faster functional recovery when the PhA was high.

PhA is a BIA parameter that has been frequently applied in clinical practice in recent years. PhA is a composite measure of tissue resistance and reactance, reflecting both soft tissue quality and quantity [35]; an increase in PhA reflects improved structural membrane integrity and cellular function, whereas a decrease in PhA indicates structural damage to the cell. The PhA of pure cell membranes is 90°, while the PhA of pure electrolyte water is 0°. In healthy individuals, PhA usually ranges from 8° to 15° [35]. In previous studies, PhA in healthy individuals peaked between the ages of 20 and 40 and then decreased significantly with age [35]. The decrease in PhA with aging may reflect cellular function and general health status in addition to body composition [36]. PhA in the present study was lower than in previous studies, possibly due to the reduced cellular function and general health status of fracture patients compared to healthy older adults.

### 4.3. Muscle Quality

Muscle quality is becoming known as an indicator for evaluating muscle. In patients with stroke, skeletal muscle mass should be considered along with intramuscular fat. Furthermore, muscle quality decreases as the intermuscular fat increases.

In addition, PhA is associated with muscle strength [37,38], for instance, PhA is higher in athletes than in non-athletes [39] and decreases with age. PhA is reduced in acute illness, inflammation, malnutrition, and prolonged inactivity [40]. It is also associated with poor quality of life [33] and poor prognosis in many different chronic diseases [41,42,43]. In the elderly, PhA is also an independent predictor of adverse clinical outcomes, such as frailty [44], falls [45], disability [46], and death [47,48]. In our previous study, we found that PhA can predict ADL recovery in elderly stroke patients [49]. The EWGSOP 2019 consensus on sarcopenia suggested that PhA can be considered an indicator of global muscle quality [14].

Muscle weakness is often seen in the elderly. This is caused by progressive congenital damage of the neuromuscular connections and impaired neuronal trophic function, resulting in random loss of muscle fibers and consequent reduction in the size of motor units [50]. Nevertheless, rehabilitation can increase muscle strength and muscle activation (neurogenic factor) in the elderly [50]. Our study showed that rehabilitation after fracture restores muscle strength and muscle quality. While some studies have suggested that rehabilitation after fracture restores muscle strength, this is the first report to show that muscle quality is involved in muscle recovery. Conversely, SMI decreased at week 4, but the difference was not statistically significant. Although this result may seem surprising, Scott et al. stated that the presence of water, cells, and adipocytes in muscle tissue is more common in older than in younger healthy subjects [51]. Such presence may reduce muscle quality [52]. In the elderly, rehabilitation may reduce water, interstitial cells, and fat in muscle, resulting in reduced muscle mass and improved muscle quality.

Furthermore, muscle strength had a correlation with ADL recovery, which is in agreement with several previous studies [7,8]. Improvement in muscle quality also correlated with ADL recovery; this finding is entirely plausible because muscle quality improvement correlated with muscle strength improvement.

Meanwhile, improvement in nutritional status did not correlate with ADL recovery. Thus, providing nutritional management to hospitalized patients with malnutrition may improve their nutritional status but not their ADL within 4 weeks. In patients with fractures, improving their nutritional status should be prioritized first before improving their ADL (after 4 weeks).

### 4.4. Study Limitations

This study has several limitations that should be considered when referring to the results.

Race and age can cause significant differences in PhA values, and the PhA values obtained in this study were smaller than in previous studies; smaller PhA values may increase the risk of developing sarcopenia, which can lead to muscle weakness. In addition, considering the nature of PFF, the number of male patients was small. Therefore, the data of male patients may not be very reliable. Future studies should include a greater number of male patients.

## 5. Conclusions

The higher the nutritional status and muscle quality of PFF patients, the quicker the recovery of ADLs. Among patients undergoing fracture rehabilitation, both muscle strength and muscle quality correlated with recovery of ADL.; SMI is not a marker of ADL recovery in elderly PFF patients. SMI may decrease with rehabilitation; therefore, using SMI as an indicator of rehabilitation effectiveness in the elderly with PFF may be inappropriate. Nutritional status, muscle strength, and muscle quality should be emphasized in the rehabilitation of these patients.

## Figures and Tables

**Figure 1 nutrients-14-02298-f001:**
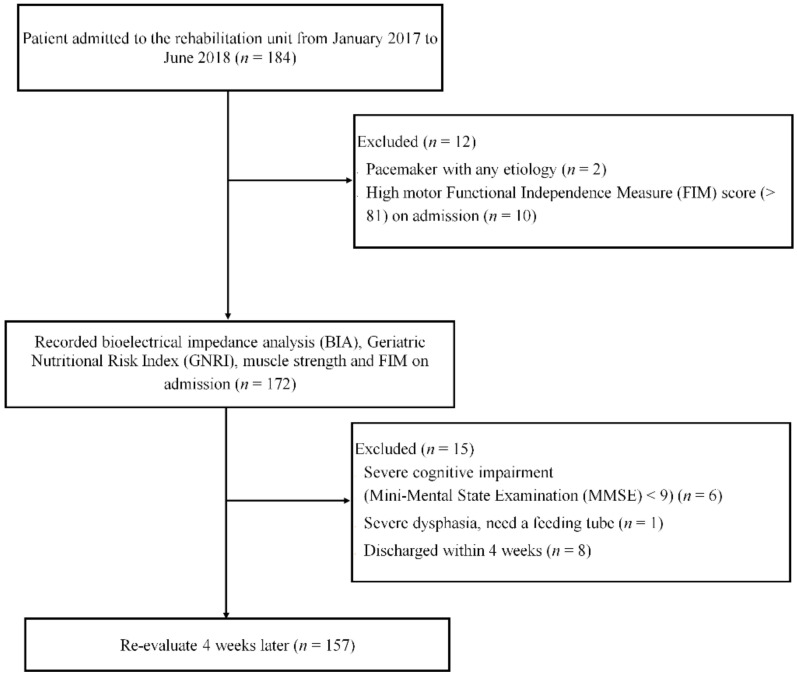
Flowchart showing patients included and excluded from the study. Initially 184 patients were enrolled, and finally 157 were evaluated.

**Figure 2 nutrients-14-02298-f002:**
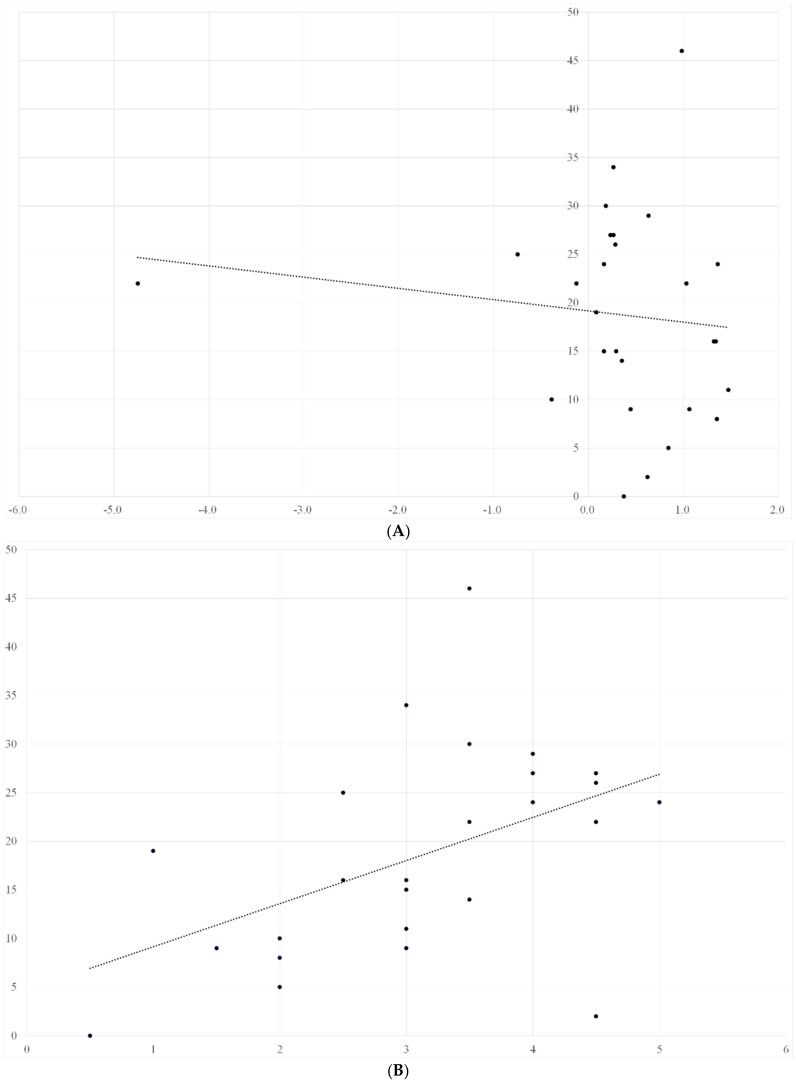
(**A**) Relationship between changes in nutritional status and ADL recovery in male patients; (**B**) relationship between changes in muscle strength and ADL recovery in male patients; (**C**) relationship between changes in muscle quality and ADL recovery in male patients; (**D**) relationship between changes in SMI and ADL recovery in male patients.

**Figure 3 nutrients-14-02298-f003:**
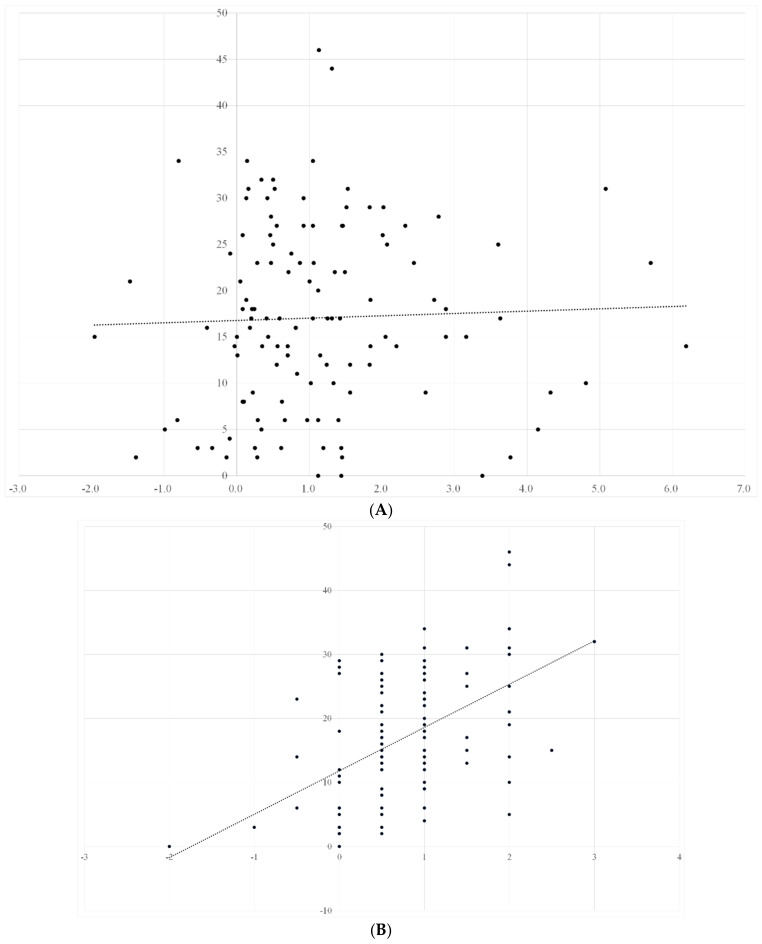
(**A**) Relationship between changes in nutritional status and ADL recovery in female patients; (**B**) relationship between changes in muscle strength and ADL recovery in female patients; (**C**) relationship between changes in muscle quality and ADL recovery in female patients; (**D**) relationship between changes in SMI and ADL recovery in female patients.

**Table 1 nutrients-14-02298-t001:** Characteristics of the study population.

Characteristics	Mean	SD
Number of patients	148	
Age (years)	84.1	7.8
Sex (F/M)	121/27	
Femoral neck fracture	84	
Trochanteric fracture	64	
BHA	76	
Osteosynthesis	72	
Days after PFF onset	22.6	8.7
Rehabilitation Program Time (min/day)	139.8	18.6
Serum albumin (g/dL)	3.6	0.4
GNRI on admission	94.3	10.9
GNRI at 4 weeks	96.8	12.8
Motor FIM score on admission	49.4	13.7
Motor FIM score at 4 weeks	66.8	16.4

BHA, bipolar hip arthroplasty; F, female; FIM, Functional Independence Measure; GNRI, geriatric nutritional risk index; M, male; PFF, proximal femur fracture; SD, standard deviation.

**Table 2 nutrients-14-02298-t002:** Characteristics of the study population according to sex.

	Male (*n* = 27)		Female (*n* = 121)	
	Mean	SD	Mean	SD
Age (years)	82.6	9.3	84.7	9.4
Height (cm)	162.1	12.8	154.2 *	9.2
Weight (kg)	53.6	14.3	48.9 *	9.8
BMI (kg/m^2^)	20.4	3.8	20.6	3.9

* *p* < 0.05; BMI, body mass index; SD, standard deviation.

**Table 3 nutrients-14-02298-t003:** Changes in nutritional status, muscle strength, quality, and SMI in male patients.

	Nutritional Status (GNRI)	Muscle Strength (kgw)	Muscle Quality (Degree)	SMI (kg/m2)	Motor FIM Items
On admission	93.6	24.7	4.2	7.4	47.6
After 4 weeks	95.8	27.9	4.4	7.3	66.4
*p*	<0.001	<0.001	<0.001	0.21	<0.001

FIM, Functional Independence Measure; SMI, skeletal muscle mass index.

**Table 4 nutrients-14-02298-t004:** Changes in nutritional status, muscle strength, quality, and SMI in female patients.

	Nutritional Status (GNRI)	Muscle Strength (kgw)	Muscle Quality (Degree)	SMI (kg/m^2^)	Motor FIM Items
On admission	94.6	18.2	3.3	5.7	49.8
After 4 weeks	95.7	19.1	3.4	5.6	66.9
*p*	<0.001	<0.001	<0.001	0.24	<0.001

FIM, Functional Independence Measure; SMI, skeletal muscle mass index.

**Table 5 nutrients-14-02298-t005:** Associations between functional recovery and clinical covariates.

Variables	Odds Ratios	95% CI	*p*
No malnutrition (GNRI > 92)	3.917	1.224–4.745	0.02
High muscle strength (male > 26.0 kgw, female > 18.0 kgw)	1.340	0.651–2.758	0.43
High muscle quality (male > 3.5°, female > 3.0°)	7.929	3.047–20.589	<0.01
High SMI (male > 7.0 kg/m^2^, female > 5.7 kg/m^2^)	0.859	0.417–1.773	0.68

CI, confidence interval; FIM, Functional Independence Measure; GNRI, geriatric nutritional risk index; SMI, skeletal muscle mass index.

## Data Availability

The data that support the findings of this study are available from the corresponding author, [H.I.], upon reasonable request.

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
