# Peer review of "Relationship between Nutritional Status, Body Composition, Muscle Strength, and Functional Recovery in Patients with Proximal Femur Fracture"

_nutrients, 2022, doi:10.3390/nu14112298_

Round 1

Reviewer 1 Report

The topic of sarcopenia, frailty and nutrition in the elderly is of sure interest. Thus the relationships between the improvement of ADL, following recovery form PFF, and  muscle quality/strength and nutrion is a relevant issue, also considering the progressive aging of our societies.

Overall the manuscript has a good layout. I add some comments/suggestions:

- the authors could mention in the methods if all the measurements were performed on the day of admission  and 4 weeks later

- did the authors evaluate the GNRI only at admission or also after rehabilitation? this could reveal if an improvement in nutritional status was effectively going on.

- was the PhA associated to GNRI?

- was the diet controlled and consistent for all the patients?

In the discussion a relevant weigh is given to the relation between nutritional status and functional recovery, however the author did not detect any raltion between ADL recovery and nutritional status. Is there any other functional parameter that could highlight a relation between recovery and nutrition?

Author Response

For Reviewer 1

We greatly appreciate your peer review.

We discussed your peer review well with our team and revised our paper.

- the authors could mention in the methods if all the measurements were performed on the day of admission and 4 weeks later

Thank you for pointing this out. We have added the following text to line 107

FIM, BIA, nutritional status and muscle strength assessments were performed on all pa-tients on the day of admission and 4 weeks later.

- did the authors evaluate the GNRI only at admission or also after rehabilitation? this could reveal if an improvement in nutritional status was effectively going on.

Thank you for pointing this out. As noted above, we assess nutritional status on admission and after 4 weeks. The results of changes in nutritional status are presented in Tables 3 and 4.

- was the PhA associated to GNRI?

Thank you for pointing this out. The current study did not examine the relationship between PhA and GNRI because this study was an investigation of factors that influence functional recovery. This will be the subject of our next study.

- was the diet controlled and consistent for all the patients?

Thank you for pointing this out. I have revised and included our dietary management policy for our patients on line 67.

According to Wakabayashi [16], the meal consumption of each patient during hospitalization was 1500–2000 kcal (protein, 1.5 g/kg/day) and was managed by the dietitian.

- In the discussion a relevant weigh is given to the relation between nutritional status and functional recovery, however the author did not detect any raltion between ADL recovery and nutritional status. Is there any other functional parameter that could highlight a relation between recovery and nutrition?

Thank you for pointing this out. As you pointed out, the association between improvement in nutritional status and functional recovery was weak.Our hypothesis was that in malnourished patients, energy would first be devoted to improving nutritional status, and then functional recovery would occur.

The discussion is described in line 295.

Reviewer 2 Report

The study conducted by Irisawa and Mizushima is interesting and presents important findings. There are a few minor concerns that the authors may want to consider before the manuscript can be published.

Introduction: This section must be improved by adding a more convincing argument. Particularly pertaining to nutritional status. Currently, only the muscle strength argument made in the introduction is strong and information related to nutritional status is not convincing enough.

I think it would be worth adding the following article to the introduction: https://www.cambridge.org/core/journals/ageing-and-society/article/abs/effects-of-a-resistance-training-community-programme-in-older-adults/F7A85415C2C7A96A34BF6633D97307CA

Methods: Did the authors register the study on ClinicalTrials.Gov similar version of the database in Japan? If yes, please provide the trial registry number.

The authors should consider adding a Consort Flow chart for the recruitment.

For BIA, it is better to provide a reference that suggests that the BIA is an alternative and validated option in relation to DXA. 

No issues with statistical analyses.

Results: 

Correlation graphs must be redesigned for clarity. Particularly the ones with yellow color. Consider using black color instead of light colors that are hard to view.

Discussion: 

This section is well written. However, it would be best to add a paragraph on the clinical implications of these findings.

Author Response

For Reviewer 2
We greatly appreciate your peer review.
We discussed your peer review well with our team and revised our paper.

-Introduction: This section must be improved by adding a more convincing argument. Particularly pertaining to nutritional status. Currently, only the muscle strength argument made in the introduction is strong and information related to nutritional status is not convincing enough.
I think it would be worth adding the following article to the introduction: https://www.cambridge.org/core/journals/ageing-and-society/article/abs/effects-of-a-resistance-training-community-programme-in-older-adults/F7A85415C2C7A96A34BF6633D97307CA
Thank you for pointing this out. We have added a sentence to the Introduction (49 lines) and added the reference you provided.

-Methods: Did the authors register the study on ClinicalTrials.Gov similar version of the database in Japan? If yes, please provide the trial registry number.
Thank you for pointing this out. This study was not registered in the Clinical Research Registry.

-The authors should consider adding a Consort Flow chart for the recruitment.
Thank you for your suggestion, We have added the flowchart as figure1.

-For BIA, it is better to provide a reference that suggests that the BIA is an alternative and validated option in relation to DXA. 
Thank you for pointing this out, I thought it should be discussed in the DISCUSSION section and added it on line 254 along with the references.

-Correlation graphs must be redesigned for clarity. Particularly the ones with yellow color. Consider using black color instead of light colors that are hard to view.
Thank you for pointing this out, I thought it should be discussed in the DISCUSSION section and added it on line 254 along with the references.

-This section is well written. However, it would be best to add a paragraph on the clinical implications of these findings.
Thank you for pointing this out. We have divided the discussion into several sections.